# Continual Learning using the SHDL Framework with Skewed Replay Distributions

## Abstract

Human and animals continuously acquire, adapt as well as transfer knowledge throughout their lifespan. The ability to learn continuously is crucial for the effective functioning of agents interacting with the real world and processing continuous streams of information. Continuous learning has been a long-standing challenge for neural networks as the repeated acquisition of information from non-uniform data distributions generally lead to catastrophic forgetting or interference. This work proposes a modular architecture capable of continuous acquisition of tasks while averting catastrophic forgetting. Specifically, our contributions are: (i) Efficient Architecture: a modular architecture emulating the visual cortex that can learn meaningful representations with limited labelled examples, (ii) Knowledge Retention: retention of learned knowledge via limited replay of past experiences, (iii) Forward Transfer: efficient and relatively faster learning on new tasks, and (iv) Naturally Skewed Distributions: The learning in the above-mentioned claims is performed on non-uniform data distributions which better represent the natural statistics of our ongoing experience. Several experiments that substantiate the above-mentioned claims are demonstrated on the CIFAR-100 dataset.

## 1 Introduction

Agents interacting with multiple tasks within any environment are required to acquire, fine-tune, and transfer knowledge for optimal functioning. The current state-of-the-art deep agents can learn and retain knowledge well when trained on individual tasks but their performance on previously learned tasks abruptly degrades when trained for one or more new tasks. This phenomenon is called catastrophic forgetting and is a long-standing challenge for machine learning and neural networks and, consequently, for the development of artificial intelligence (AI) systems (Hassabis et al. (2017); Thrun & Mitchell (1995)).

McCloskey & Cohen (1989) suggested the underlying cause of forgetting to be the distributed shared representation of tasks via network weights. Therefore training the network towards a new objective can cause almost complete forgetting of former knowledge. Several techniques have been proposed to encode knowledge in non-overlapping representations or; preserve or limit changes to the weights corresponding to the learned knowledge.

Kortge (1990) attempted to encode knowledge for different tasks with minimal overlap via activation sharpening algorithms. Lew recorded the inputs orthogonal and attempted orthogonal encoding at all hidden layers (McRae & Hetherington (1993), French (1994)). Recently, encoding methods like maxout and dropout (Goodfellow et al. (2013)) and local winner-takes-all (Srivastava et al. (2013)) have been explored to create sparsified feature representations to minimize task-specific representation overlap.

However, mammalian brains undergo only gradual systematic forgetting suggesting that shared representations may not be the root cause of the problem. Several recent approaches have developed strategies that slow down learning on network weights which are important for previously learnt tasks. Kirkpatrick et al. (2016) preserved weights which correlate with previously acquired knowledge by using a fisher information matrix based regularizer. Zenke et al. (2017) slowed down learned on weights important for the previous tasks by using path integrals of loss-derivatives. An extreme version of the above-mentioned methods are Progressive neural networks (Rusu et al. (2016)) and Pathnets (Fernando et al. (2017)) that directly freeze important pathways in neural networks. This

eliminates forgetting altogether but requires growing the network after each task and can cause the architecture complexity to grow with the number of tasks. These methods give superior performance as compared to the sparse representations encoding methods (detailed before) but may not be explicitly targeting the cause of catastrophic forgetting.

Neuroscientific evidence (McClelland et al. (1995), O'Neill et al. (2010)) has inspired another set of methods which focus on the replay of previously seen samples along with new samples to avoid catastrophic forgetting. The replay methods usually maintain an episodic memory of fixed-size with exemplars which are either directly replayed while learning e.g. in iCaRL (Rebuffi et al. (2017)) or indirectly utilized to adapt future gradient updates to the network e.g. inGEM (Lopez-Paz & Ranzaton (2017)) to avoid forgetting on previously seen tasks. Robins, 2004 proposed another replay technique, known as the pseudo-pattern rehearsal, to preserve the learned mappings by uniformly sampling random inputs and their corresponding outputs from networks and replaying them along with new task samples.

Despite the superior performance of these methods and their biological significance, choosing to store samples from previous tasks is challenging as a large working memory may be required (Lucic et al. (2017)). Recently, few generative approaches have been proposed to overcome this issue as these methods allow for storage of the relevant samples from previous tasks with limited memory. However, these methods give inferior performance as compared to methods with large storage capacity.

This paper proposes a training regime that uses the modular ScatterNet Hybrid Deep Learning (SHDL) network proposed by Singh et al (Singh & Kingsbury (2017a; 2018)) to achieve lifelong learning over two phases. The proposed methodology can retain the learned knowledge over phase 1 with minimal forgetting while using this learned knowledge to rapidly learn (forward transfer) new knowledge over phase 2. The architecture of the modular network is computationally efficient and can learn the relevant representations using relatively fewer labelled examples as compared to the end-to-end supervised deep networks.

The contributions of the paper are as follows:

- ***Modular Architecture***: This work uses the modular ScatterNet Hybrid Deep Learning (SHDL) network proposed by Singh et al. (Singh & Kingsbury (2017a; 2018)) which is composed of a handcrafted frontend, an unsupervised midsection, and a supervised backend. The network is computationally efficient architecture and is constructed by selecting the optimal number of filters in each layer of the network. The network can learn rapidly with fewer labelled examples. This is beneficial as it is difficult to obtain a large number of labelled examples for most applications.

- ***Rapid Learning with Forward Transfer***: In addition to the ability of the proposed training regime to retrain previously learned knowledge with minimal forgetting. The framework is also able to build on the previously learned knowledge to acquire new knowledge rapidly leading to faster learning.

- ***Efficient Memory Storage with Naturally Skewed Distributions***: Most of the work on lifelong learning has primarily assumed iid distribution for all classes during replay. This is less likely to be true for the human experience as the distribution of classes seems to be more skewed towards the current experience over the recent ones. This distribution of data replayed to the network in this work follows similar natural assumptions. This also allows for efficient memory storage as only a fraction of data for the previous tasks needs to be stored.

The classification performance of the proposed training regime is evaluated on the CIFAR-100 dataset for two phases. Multiple experiments are conducted with different relative replay frequencies between the learned tasks (phase-0) and the new tasks (phase-1).

Section 2 of the paper briefly presents the proposed training regime along with the SHDL network. Section 3 presents the experimental results while Section 4 draws conclusions.

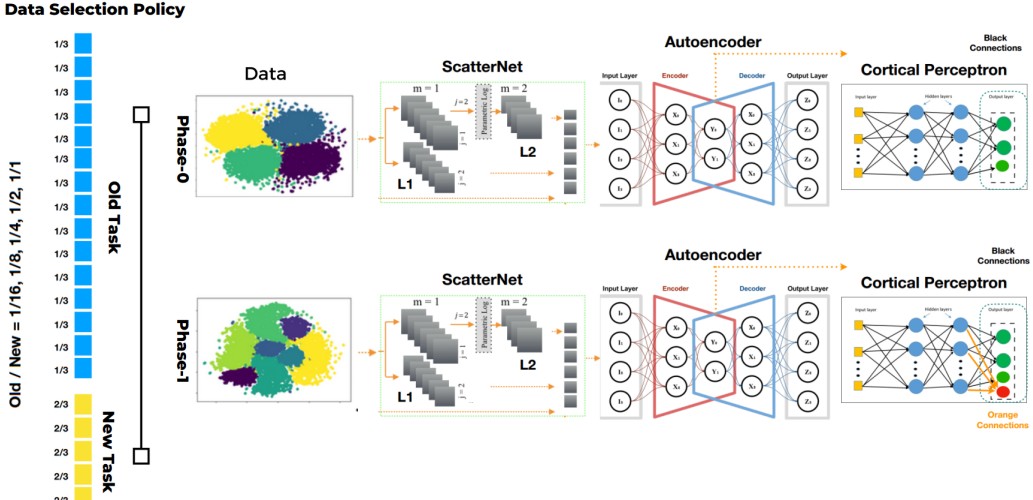

Figure 1: The illustration presents the training regime used for continuous learning over two phases with the SHDL framework composed of the handcrafted frontend, unsupervised midsection, and supervised backend. In phase-0, the network learns an initial set of 15 classes (green connections). In phase-1, 5 new classes are introduced to the network while continuing training of the old 15 classes at different relative frequencies (0.0625, 0.125, 0.25, 0.50, 1.0)

## 2 TRAINING REGIME WITH SCATTERNET HYBRID DEEP LEARNING (SHDL) FRAMEWORK

This section introduces the proposed training regime for continual learning with the ScatterNet Hybrid Deep Learning (SHDL) Framework over two phases. The section presents the mathematical formulation of SHDL framework along with the training details (Fig. 1).

### 2.1 SCATTERNET HYBRID DEEP LEARNING (SHDL) FRAMEWORK

The ScatterNet Hybrid Deep Learning (SHDL) Framework is composed of a handcrafted frontend, an unsupervised midsection, and a supervised backend. Each layer of the network is designed and optimized automatically that produces the desired computationally efficient architecture. The handcrafted features allow for the rapid learning of the unsupervised module as it is not necessary to wait for the first layer to learn edges as they are already present. The unsupervised learning module is quite advantageous as it allows the network to learn meaningful representation from relatively fewer labelled examples as compared to fully supervised networks (Singh & Kingsbury (2017a; 2018)).

#### 2.1.1 HAND-CRAFTED MODULE: SCATTERNET

Handcrafted descriptors are extracted using the parametric log based two-level Dual-Tree Complex Wavelet Transform (DTCWT) ScatterNet ( Singh & Kingsbury (2017b)) that extracts relatively symmetric translation invariant low-level features at the first layer and more discriminative sparse features at second layer (Bruna & Mallat (2013); Singh & Kingsbury (2016)). The extracted features are also dense over the scale as they are obtained by decomposing multi-resolution images obtained at 1.5 times (R1) and twice (R2) the size of the input image. This ScatterNet (Singh & Kingsbury (2017b)) is chosen over Bruna and Mallat's (Bruna & Mallat (2013)) due to its superior classification accuracy and computational efficiency. The DTCWT ScatterNet formulation is presented for an input signal ($x$) which may then be applied to each multi-resolution image.

The features at the first layer are obtained by filtering the input signal $x$ with dual-tree complex wavelets $\psi_{j,r}$ at different scales ($j$) and six pre-defined orientations ($r$) fixed to $15°, 45°, 75°, 105°, 135°$ and $165°$, as shown in Fig. 2. A more translation invariant representa-

tion is built by applying a point-wise $L_2$ non-linearity (complex modulus) to the real and imaginary part of the filtered signal:

$$U[L1] = \sqrt{|x \star \psi_{\lambda_1}^a|^2 + |x \star \psi_{\lambda_1}^b|^2} \qquad (1)$$

The parametric log transformation layer is applied on the oriented features, extracted at the first scale $j = 1$ with a parameter $k_{L1}[j]$, to reduce the effect of outliers by introducing relative symmetry (rs) to their amplitude distribution (as shown in Fig. 3):

$$U1_{rs}[j] = \log(U[L1][j] + k_{L1}[j]), \quad U[L1][j] = |x \star \psi_j|, \qquad (2)$$

The parameter $k_{L1}$ is selected such that it minimizes the difference between the mean and median of the distribution Singh & Kingsbury (2017b). Next, a local average is computed on the envelope $|U1[\lambda_{L1}]|$ that aggregates the coefficients to generate the desired translation-invariant representation:

$$S_{rs}[L1] = |U1_{rs}| \star \phi_{2^J} \qquad (3)$$

The energy (high-frequency components) lost due to smoothing is recovered by cascaded wavelet filtering applied at the second layer Bruna & Mallat (2013). The recovered components are again not translation invariant so invariance is achieved by first applying the $L_2$ non-linearity to obtain the regular envelope:

$$U[L2] = |U1_{rs} \star \psi_{\lambda_{L2}}| \qquad (4)$$

The parametric log transformation is applied again to produce relative symmetry:

$$U2_{rs}[j] = \log(U[L2][j] + k_{L2}[j]) \qquad (5)$$

Next, a local-smoothing operator is applied to improve translation invariance::

$$S_{rs}[L2] = U2_{rs} \star \phi_{2^J} \qquad (6)$$

The output coefficients are typically formed from $x \star \phi$ (Layer 0), $S_{rs}[L1]$ (Layer 1) and $S_{rs}[L2]$ (Layer 2) for each of the two image resolutions R1 and R2.

### 2.1.2 UNSUPERVISED LEARNING MODULE: CONVOLUTIONAL AUTOENCODER

The Scattering features extracted at (L0, L1, L2) are concatenated and given as input to 4 stacked convolutional Autoencoder (MasciUeli et al. (2011)). An autoencoder is a neural network that is trained to copy its input to its output. Internally, it has a hidden layer $h$ that describes a code used to represent the input. The network may be viewed as consisting of two parts: an encoder function $h = f(x)$ and a decoder that produces a reconstruction $r = g(h)$.

The autoencoder learns useful information in the hidden layer $h$ as it is constrained to have a smaller dimension than $x$ (undercomplete). Learning an undercomplete representation forces the autoencoder to capture the most salient features of the training data.

The learning process is described simply as minimizing a loss function

$$L_{auto}(x, g(f(x))) \qquad (7)$$

where $L_{auto}$ is a loss function penalizing $g(f(x))$ for being dissimilar from $x$, such as the mean squared error.

### 2.1.3 SUPERVISED LEARNING MODULE: PERCEPTRON

The features learned by the hidden layer of the autoencoder are used by a perceptron (Rosenblatt (1958)) to learn high-level features. The perceptron is trained using the binary cross entropy (BCE) with logits that is defined as:

$$L_{perc} = -\frac{1}{N} \sum_{i=0}^{N-1} (y_i \log \sigma(\hat{y}_i) + (1 - y_i) \log(1 - \log \sigma(\hat{y}_i))) \tag{8}$$

where $L_{perc}$ is a loss function computed on $N$ samples with $y$ as the ground truth label and $\hat{y}$ is the prediction of the output layer of the perceptron.

## 2.2 PROPOSED TRAINING REGIME FOR CONTINUAL LEARNING

The proposed training regime is used with the SHDL framework for continual learning over two phases. In the first phase (Phase-0), the SHDL framework is trained with the initial set of classes while in the second phase (Phase-1), the framework is trained with new classes and only a fraction of items from the Phase-0 classes. Since only a fraction of data is required to be stored, this leads to efficient memory storage. Also, knowledge learned during the first phase aids the learning of phase-1 classes.

Only the autoencoder and percepton modules of the framework are trained as the front-end is hand-crafted. The autoencoder is jointly trained from scratch for classes of both phases to learn mid-level features. The module is trained on the scatternet features and is not evolved once it is trained. The percepton, on the other hand, is trained in two phases. In Phase-0, the output layer of the perceptron is composed of nodes (green) equivalent to the number of classes and initialized randomly, as shown in Fig. 1. In phase-1, new classes are introduced to the framework corresponding to which new nodes (orange) are added in the output layer and again initialised randomly. However, the weights of all three layers of the perceptron, learned for the phase-1 classes are carried forward. The phase-1 training is performed with new classes along with a fraction of items from the classes introduced in phase-0. The weights carried forward for the first phase (Phase-0) allows the network to retain the performance on them even with the only fraction of the samples. The perceptron is trained using the BCE with logits as stated above.

## 2.3 REPLAY WITH RELATIVE FREQUENCIES

The data distribution introduced to the SHDL framework in phase-1 is skewed towards the data samples for the new classes. This skewed distribution is constructed by sampling the items for the known classes relative to the new classes using the replay parameter ($S_p$) that is defined below:

$$R_p = \frac{Known_{items}}{New_{items}} \tag{9}$$

where $Known_{items}$ corresponds to the items of the known (phase-0) classes while $New_{items}$ are items of the new (phase-1) classes. As an example, a 0.5 sampling parameter would correspond to 1 old item per 2 new items in the distribution.

## 2.4 PERFORMANCE MEASURES FOR CONTINUAL LEARNING

The performance of the continual learning system is measured on the ability of the system to effectively learn new knowledge (phase-1) while being able to retain the previously learned knowledge (phase-0).

The performance on the new knowledge is measured using the (i) final performance and, (ii) the rate of learning while the retention on the learned knowledge in phase-0 is measured on the (iii) final performance and, (iv) time to final performance. The measures for the both learned and new knowledge are detailed below:

The performance measures for the phase-1 knowledge acquired by the network are detailed below:

- **Performance gained**: The system should reach similar or higher asymptotic performance compared to the baseline for new (phase-1) classes. This is measured as detailed below:

$$P_g = \frac{Acc_{phase-1(new)}}{Acc_{baseline}} \tag{10}$$

where $P_g$ corresponds to the performance gained ($P_g > 1$) or lost ($P_g < 1$) by the system for both new or known classes. For the ideal case, $P_g$ should be greater then or equal to one.

- **Rate of learning**: The knowledge learned during phase-0 should ideally aid the learning of phase-1 classes. This is measured using slope gained and area gained as detailed below:

  - **Slope Gained**: The slope gained represents the faster acquisition of new knowledge is measured as detailed below:

  $$S_g = \frac{S_{phase-1(new)}}{S_{baseline}} \tag{11}$$

  where $S_g$ represents the slope gained. $S$ corresponds to the slope of the learning curves obtained by fitting a sigmoid curve (dotted black curve) to them as shown in Fig. 2. $R > 1$ represents faster learning as compared to the baseline.

  - **Area Gained**: Area gained by the phase-1 curve relative to the baseline is also used to measure the faster acquisition of new knowledge as shown below:

  $$A_g = \frac{A_{phase-1(new)}}{A_{baseline}} \tag{12}$$

  where $A_g$ corresponds to the area gained until convergence. $A_{phase-1(new)}$ represents the area under the curve for the new classes while $A_{baseline}$ represents area under the curve for the baseline curve. $A_g > 1$ would represents positive area gained representing faster learning relative to the baseline.

The performance measures for the retention of phase-0 knowledge are detailed below:

- **Performance gained**: The system should reach similar close to the asymptotic baseline performance and is measure as shown below:

$$P_g = \frac{Acc_{phase-1(old)}}{Acc_{baseline}} \tag{13}$$

where $P_g$ corresponds to the performance gained ($P_g > 1$) or lost ($P_g < 1$) by the system. For the ideal case, $P_g$ should be greater then or equal to one.

- **Time to final Performance**: The performance of the learned knowledge during phase-0 gets interfered when new knowledge is introduced to the network. The performance on the learned tasks is gradually restored to a new asymptotic performance. The time to final performance ($TFP_{old}$) should ideally be immediate so smaller the time to the restoration better is the model.

## 3 EXPERIMENTAL RESULTS

The performance of the proposed training regime is evaluated on the SHDL framework with several experiments conducted on the CIFAR-100 dataset (Krizhevsky & Hinton (2009)). The experiments are constructed based on the relative frequencies between the new and known classes as well as the nature of the nature classes. The performance is measured based on the measures defined as detailed above. The details of the experiments as well as the training details of each section are presented below.

### 3.1 EXPERIMENTS ON CIFAR-100 DATASET

The CIFAR-100 (Krizhevsky & Hinton (2009)) dataset is constructed of 100 classes which are composed of 20 classes with each class further divided into 5 sub-classes. Each class consists of 500 training and 100 test images with each image of size $32 \times 32 \times 3$.

The training regime as detailed consists of phase-0 where 15 classes sampled from the CIFAR-100 are introduced to the SHDL framework. In the second phase, 5 new classes are presented to the framework.

Table 1: Classification error (%) on the CIFAR-10 dataset for the original CNN architectures and their corresponding DTSCNN architectures.

| $S_p$ | Near Category | | | | | Far Category | | | | |
|---|---|---|---|---|---|---|---|---|---|---|
| | New | | | Old | | New | | | Old | |
| Set-1 | $S_g$ | $A_g$ | $P_{new}$ | $P_{old}$ | $TFP_{old}$ | $S_g$ | $A_g$ | $P_{new}$ | $P_{old}$ | $TFP_{old}$ |
| 0.062 | 1.87 | 1.98 | 1.38 | 0.63 | 1356 | 3.68 | 2.05 | 2.42 | 0.58 | 1348 |
| 0.125 | 1.64 | 1.76 | 1.33 | 0.77 | 1210 | 3.56 | 1.86 | 2.36 | 0.71 | 1121 |
| 0.250 | 1.17 | 1.23 | 1.24 | 0.92 | 1008 | 2.91 | 1.41 | 2.24 | 0.86 | 865 |
| 0.500 | 0.82 | 1.09 | 1.12 | 1.03 | 656 | 1.05 | 1.21 | 1.99 | 1.00 | 512 |
| 1.000 | 0.38 | 0.96 | 0.94 | 1.11 | 316 | 0.94 | 0.98 | 1.67 | 1.12 | 449 |
| Set-2 | $S_g$ | $A_g$ | $P_{new}$ | $P_{old}$ | $TFP_{old}$ | $S_g$ | $A_g$ | $P_{new}$ | $P_{old}$ | $TFP_{old}$ |
| 0.062 | 1.83 | 1.90 | 1.44 | 0.68 | 1248 | 3.53 | 2.13 | 2.53 | 0.63 | 1367 |
| 0.125 | 1.60 | 1.68 | 1.39 | 0.77 | 1215 | 3.19 | 1.92 | 2.41 | 0.79 | 1148 |
| 0.250 | 1.13 | 1.15 | 1.30 | 1.44 | 1108 | 2.781 | 1.54 | 2.33 | 0.94 | 889 |
| 0.500 | 0.78 | 1.01 | 1.18 | 1.58 | 766 | 1.05 | 1.29 | 1.99 | 1.16 | 512 |
| 1.000 | 0.34 | 0.88 | 0.99 | 0.27 | 343 | 0.77 | 1.03 | 1.74 | 1.24 | 452 |
| Set-3 | $S_g$ | $A_g$ | $P_{new}$ | $P_{old}$ | $TFP_{old}$ | $S_g$ | $A_g$ | $P_{new}$ | $P_{old}$ | $TFP_{old}$ |
| 0.062 | 1.93 | 2.05 | 1.40 | 0.76 | 1287 | 3.67 | 2.19 | 2.72 | 0.49 | 1321 |
| 0.125 | 1.70 | 1.81 | 1.36 | 0.85 | 1262 | 3.51 | 1.72 | 2.59 | 0.76 | 1108 |
| 0.250 | 1.23 | 1.28 | 1.20 | 1.02 | 1178 | 2.66 | 1.32 | 2.38 | 0.81 | 796 |
| 0.500 | 0.88 | 1.14 | 1.12 | 1.13 | 796 | 0.96 | 1.17 | 1.93 | 1.09 | 541 |
| 1.000 | 0.44 | 1.01 | 0.87 | 1.25 | 436 | 0.94 | 1.09 | 1.54 | 1.27 | 423 |
| Set-4 | $S_g$ | $A_g$ | $P_g$ | $P_g$ | $TFP_{old}$ | $S_g$ | $A_g$ | $P_g$ | $P_g$ | $TFP_{old}$ |
| 0.062 | 2.04 | 1.98 | 1.34 | 0.55 | 1378 | 3.55 | 2.31 | 2.42 | 0.64 | 1375 |
| 0.125 | 1.81 | 1.76 | 1.27 | 0.67 | 1234 | 3.23 | 1.91 | 2.36 | 0.83 | 1097 |
| 0.250 | 1.34 | 1.23 | 1.19 | 0.82 | 1076 | 2.19 | 1.32 | 2.24 | 0.98 | 786 |
| 0.500 | 0.99 | 1.09 | 1.05 | 1.10 | 867 | 1.56 | 1.07 | 1.99 | 1.14 | 493 |
| 1.000 | 0.55 | 0.96 | 0.91 | 1.19 | 398 | 0.83 | 0.84 | 1.67 | 1.29 | 431 |

Phase-0 consists of 500 examples for each class while in phase-1, the items in the 15 phase-0 classes are selected with frequency relative to the 5 phase-1 classes. The relative frequencies with which the known items are sampled are 0.0625, 0.125, 0.25, 0.50, 1.0. The 0.5 frequency means that the phase-1 distribution consists of $500 \times 5$ items for 5 new classes and $250 \times 15$ items for the 15 new classes.

The full test set of 10000 images is used for evaluation on all the experiments.

Four set (set-1 to set-4) of experiments are constructed by randomly selecting 15 phase-0 classes and then the corresponding 5 phase-1 classes.

## 3.2 SHDL Training Details

The SHDL framework consists of a scatternet frontend, the autoencoder midsection and a supervised perceptron as the backend.

The frontend scatternet features are extracted from the input RGB image using DTCWT filters at 2 scales and 6 fixed orientations. Next, log transformation with parameter $k_j = 1.1$ is applied to the representations obtained at the finer scale to introduce relative symmetry (Section. 2.1).

The convolutional autoencoder is composed of 4 stacked layers with layers that learns 256, 128, 64, and 32 filters respectively. The autoencoder is trained using SGD with a learning rate of $0.001$ with a batch size of 32. The 32 feature map of size $4 \times 4$ are obtained at the hidden layers which are flattened to produce a feature vector of size 512 ($32 \times 4 \times 4$).

The 512 dimensional vector is given as input to the 3 layers perceptron with 128, 64, 15 or 20 (phase-0/phase-1) nodes per layer. The perceptron is trained to minimize the BCE loss using SGD with a learning rate of $0.001$ and a batch size of 16.

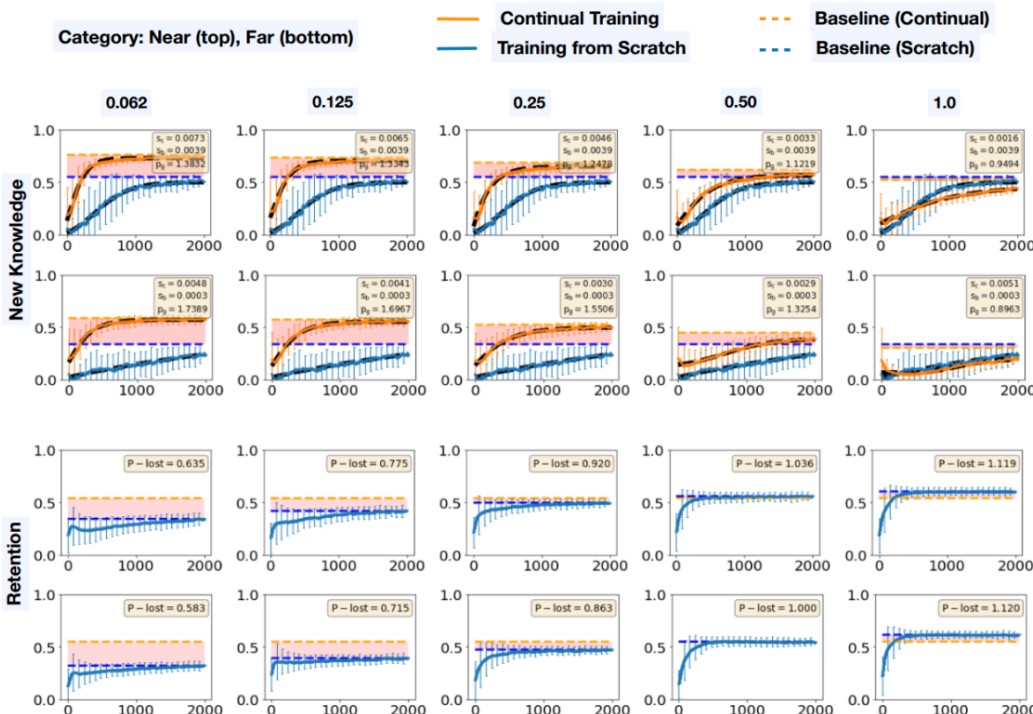

Figure 2: The illustration presents plots for new knowledge and retention for both near and far experiments. The new knowledge plots show the rapid learning of new classes during the second phase as compared to the baseline. The retention plots for the classes learned during the first phase shows retention with only transient interference. Orange curve denoted the phase-1 plots while the blue curve represents the phase-0 plots. The dotted black curve represents the sigmoid fit to curves for both phases. The x axis of the graphs represents the number of iterations. The performance measure at each iteration is obtained by averaging the performance measures over 10 batches of 16 items each.

### 3.3 EVALUATION ON THE PROPOSED MEASURES

The performance of the proposed training regime on different measures is shown in Table. 1 and Fig. 2. The x axis of the graphs in Fig. 2 represents the number of iterations. The performance measure at each iteration is obtained by averaging the performance measures over 10 batches of 16 items each.

As observed from the table, the slope ($S_g$), area ($A_g$), and performance ($P_g$) is gained for new (phase-1) knowledge (>1) for both near and far experiments corresponding to all 4 set of experiment, when a minimum of 4 new item are replayed for each old item ($R_p \leq 0.25$). The performance on the known (phase-0) knowledge is lost ($P_g < 1$) with $R_p \leq 0.25$ and requires replay for more than 1000 ($TFP_{old}$) iterations to reach final performance.

As more data samples for the learned knowledge are replayed ($R_p \geq 0.50$), the phase-0 performance is recovered much more rapidly ($TFP_{old} \leq 1000$) and with no or minimal performance loss. However, it gets much harder to learn new (phase-1) knowledge as indicated by $S_g < 1$. This is also shown in Fig. 2

Table. 1 indicates that the he ideal scenario to learn phase-1 knowledge rapidly with minimal loss to the knowledge learned during phase-0 is when 4 new item are replayed for each old item ($R_p \leq 0.25$). This also leads to efficient memory storage as fewer samples from the old classes are required to be stored.

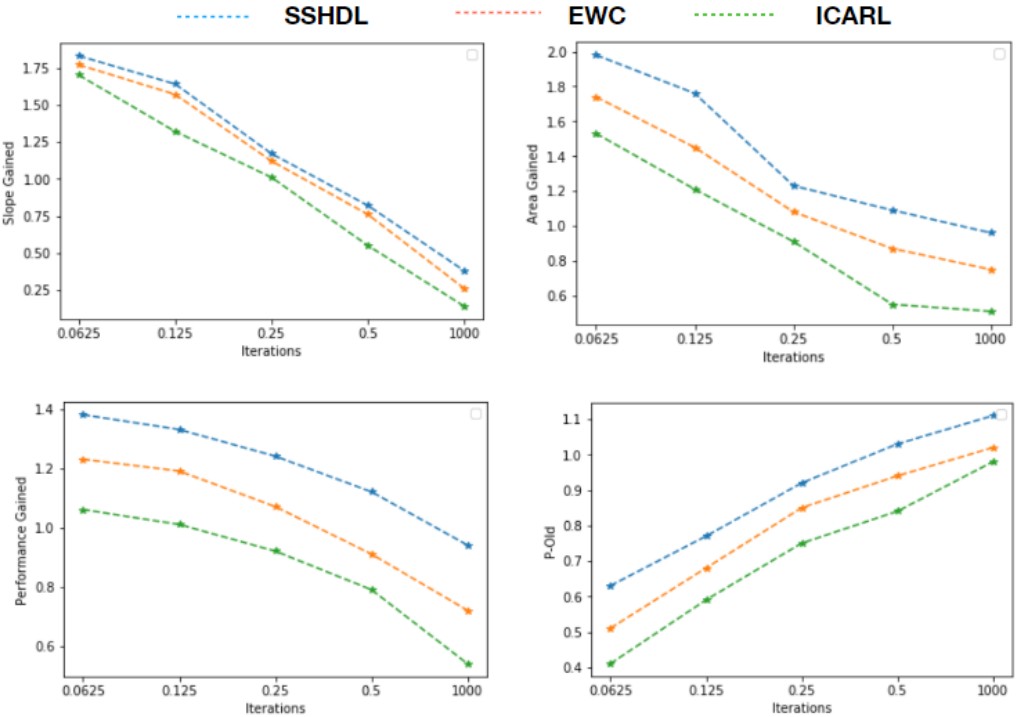

Figure 3: The illustration presents the performance of the proposed SHDL framework against SOTA EWC and ICARL frameworks- Slope gained, Area gained, Performance gained (old (phase-0), new (phase-1)).

### 3.4 COMPARISON WITH THE STATE-OF-THE-ART

The performance of the proposed training regime is compared with two state-of-the-art methods, Elastic Weight Consolidation (EWC) (Kirkpatrick et al. (2016)) and Incremental Classifier and Representation Learning (iCARL) (Rebuffi et al. (2017)).

EWC (Kirkpatrick et al. (2016)) is designed to avoid catastrophic forgetting by regulating the loss function while iCARL (Rebuffi et al. (2017)) is a class-incremental learner that classifies using a nearest-exemplar algorithm, and prevents catastrophic forgetting by using an episodic memory.

Table 2: The number of iterations required to reach final performance for old knowledge presented for the proposed and two state-of-the-art methods, EWC (Kirkpatrick et al. (2016)) and iCARL (Rebuffi et al. (2017)o

| $R_p$ | 0.0625 | 0.125 | 0.25 | 0.5 | 1.0 |
|---|---|---|---|---|---|
| SHDL | 1296 | 1169 | 1057 | 651 | 459 |
| EWC | 1418 | 1381 | 1239 | 987 | 789 |
| iCARL | 2215 | 1794 | 1611 | 1211 | 909 |

The performance of the proposed method and the state-of-the-art is evaluated on the proposed measures and shown in Fig. 3 and Table 2. The proposed method is able to outperform both EWC and iCARL.

## 4 CONCLUSION

This work proposes a modular SHDL architecture that is able to acquire new knowledge rapidly while averting catastrophic forgetting by replaying replay only a fraction of items from the previ-

ously learned knowledge learning to memory efficient architecture. The performance of the proposed method is validated on several proposed measures. The proposed training regime is able to outperform two state-of-the-art methods on the proposed measures further establishing the usefulness of the proposed method.

ACKNOWLEDGMENTS

This research project has been supported by the lifelong Learning Machines (L2M) program of the Defence Advanced Research Projects Agency (DARPA). The U.S. Government is authorized to reproduce and distribute reprints for governmental purposes notwithstanding any copyright annotation thereon. Disclaimer: The views and conclusions contained herein are those of the authors and should not be interpreted as necessarily representing the official policies or endorsements, either expressed or implied, of DARPA, or the U.S. Government.

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
