# OpenReview forum: "Continual Learning using the SHDL Framework with Skewed Replay Distributions"
_ICLR.cc/2020/Conference — Reject_

### Official Review · AnonReviewer2 · 2019-10-23
**Official Blind Review #2**

**Rating:** 1

**Review:**

The paper suggest to use the previously proposed ScatterNet Hybrid Deep Learning  (SHDL) network in a continual learning setting. This is motivated by the fact that the SHDL needs less supervised data, so keeping a small replay buffer can be enough to maintain performance while avoiding catastrophic forgetting.

My main doubt is the benchmark and evaluation of the proposed method. The metrics reported are all relative to a baseline value (which I could not find reported), and make it difficult to understand how the model is performing in absolute term. This is particularly a problem when comparing with existing state of the art method (Fig. 3, Table 4), since this does not exclude that they may have an overall much better accuracy in absolute terms.

Also concerning the comparison with the previous literature, I could find no details about the architecture and the training algorithm used. Notice that this may in particular affect some the reported metrics, since they depend on the shape of the training curve (reporting the training curves for all methods may also be useful). Also, since SHDL uses a small replay buffer, are EWC and the other method modified to use the replay buffer and make the comparison fair?

While several standard tests for continual learning exists (for example the split CIFAR10/100 in Zenke et al., 2017), those are not used, and rather a simpler test is used which only attempt to learn continually two datasets.  It would be helpful to also report a direct comparison on those tests.

Regarding the line: "The autoencoder is jointly trained from scratch for classes of both phases to learn mid-level features", does this mean that the auto-encoder is trained using data of the two distributions at the same time rather than one after the other? If it is the former case, while it is unsupervised training, it would be a deviation from the standard continual learning framework and should clearly be stated.



**Experience Assessment:**

I have published one or two papers in this area.

**Review Assessment: Checking Correctness Of Derivations And Theory:**

N/A

**Review Assessment: Checking Correctness Of Experiments:**

I carefully checked the experiments.

**Review Assessment: Thoroughness In Paper Reading:**

I read the paper at least twice and used my best judgement in assessing the paper.

---

### Official Review · AnonReviewer1 · 2019-10-27
**Official Blind Review #1**

**Rating:** 1

**Review:**

The author proposed a modular SHDL with skewed replay distributions to do continual learning and demonstrated the effectiveness of their model on the CIFAR-100 dataset. They made contributions in three aspects: (1) using a computationally efficient architecture SHDL which can learn rapidly with fewer labeled examples. (2) In addition to retain previous learned knowledge, the model is able to acquire new knowledge rapidly which leads to fast learning. (3) By adopting the naturally skewed distributions, the model has the advantage of efficient memory storing.

Overall, the paper should be rejected because
(1)the author spent too much space to introduce the off-the-shelf SHDL model which should be put in the Appendix or referred directly. In other words, the author should explain more details about the “replay” mechanism in their model and show the advantage of choosing SHDL rather than other deep neural nets under the continual learning paradigm.
(2)The comparison with other methods are too simple. When comparing with other methods, the author should introduce the parameter setting and the detailed training strategy. Otherwise, the evidence made in the experimental section is not convincing. Besides, the author should follow the evaluation paradigm used in other published papers to make a fairer comparison.
(3)The author should carry out more discussion about which part of their model contributes the most to the continual learning. After reading the paper thoroughly, I am still unclear about it.

The paper has some imprecise part, here are a few:
(1)The caption in Table 1 is too simple. More details should be add to explain the table.
(2)What is the DTSCNN in Table 1?
(3)What is the green connections in Figure 1?
(4)In the second contribution “Rapid learning with forward transfer”, is the ability to “retrain” or “retain” the previous learned knowledge?

**Experience Assessment:**

I have read many papers in this area.

**Review Assessment: Checking Correctness Of Derivations And Theory:**

I assessed the sensibility of the derivations and theory.

**Review Assessment: Checking Correctness Of Experiments:**

I assessed the sensibility of the experiments.

**Review Assessment: Thoroughness In Paper Reading:**

I read the paper thoroughly.

---

### Official Review · AnonReviewer4 · 2019-11-01
**Official Blind Review #4**

**Rating:** 1

**Review:**

I think there might be some interesting ideas in the work, but I think the authors somehow did not manage to position themselves well within the *recent* works on the topic or even with respect to what continual learning (CL) is understood to be in these recent works.

E.g. CL is a generic learning problem, and most algorithms are generic (with a few caveats in terms of what information is available) in the sense that they can be applied regardless of task (be it RL, be it sequence modelling etc.). This work seems limited to image classification. The SHDL wavelets pre-processing, if I understood it, is specific for images and probably even there under some assumption (e.g natural images).

The autoencoder (middle bit) is trained on all tasks the CL needs to face, if I understood the work correctly (phase 0 + phase 1). This potentially makes the CL problem much simpler because you are limiting yourself to the top layer only when dealing with CL, not the rest. Not to mention that I don't understand the motivation of the autoencoder. I think ample results show that unsupervised learning fails in many instances to provide the right features and underperforms compared to learning discriminative features by just backproping from the cross entropy (discriminative loss) all the way down. The only instance I know of for doing this is in low data regime where there is no alternative.

I think the modularity used needs to be better introduced. Why the autoencoder, why the first layer of wavelets? Is it for the benefit for CL? I can understand the wavelets, since they are not learnt. But the autoencoder? The autoencoder being trained on all data feels like a cheat.

I think the citation of the perceptron a bit strange. Do you really use the original perceptrion from 58? Why? We have much better tools now !?

I think the different metrics introduced are interesting and useful. Though you should somehow find common ground to existing works as well to ensure a point of comparison. In the results section I almost got lost. What is the final performance on Cifar. How does this compare to a model that is not trained in a CL regime? What loss do you get from the proposed parametrizaton?

In the comparison with EWC and iCarl, there the whole model was dealing with the CL problem, right? (all intermediary layers). I'm actually surprised iCarl is not doing better (I expect it can do better than EWC). Maybe provide a few more information of hyperparam used for this comparison.

Overall I think the paper is not ready for being published. Not without addressing these points:
 * role of modularity (if not CL -- then why? ; is the modular structure original or part of the previous works cited, e.g. where the wavelets are introduced and so forth)
 * better integration with recent literature; provide answers and settings that allow apple to apple comparison so one can easily understand where this approach falls; if the method is not meant for this "traditional settings and metrics" please still provide them, and then motivate why this regime is not interesting and explain better the regime the method is meant for
 * as it stands the work is light on the low level details; Hyper-params and other details are not carefully provided (maybe consider adding an appendix with all of these). I have doubts that the work is reproducible without these details.

**Experience Assessment:**

I have published in this field for several years.

**Review Assessment: Checking Correctness Of Derivations And Theory:**

I assessed the sensibility of the derivations and theory.

**Review Assessment: Checking Correctness Of Experiments:**

I assessed the sensibility of the experiments.

**Review Assessment: Thoroughness In Paper Reading:**

I made a quick assessment of this paper.

---

### Decision · Program_Chairs · 2019-12-19

**Decision:**

Reject

**Comment:**

The paper adapts a previously proposed modular deep network architecture (SHDL) for supervised learning in a continual learning setting.  One problem in this setting is catastrophic forgetting.  The proposed solution replays a small fraction of the data from old tasks to avoid forgetting, on top of a modular architecture that facilitates fast transfer when new tasks are added.  The method is developed for image inputs and evaluated experimentally on CIFAR-100.

The reviews were in agreement that this paper is not ready for publication.  All the reviews had concerns about the lack of explanation of the proposed solution and the experimental methods.  The reviewers were concerned about the choice of metrics not being comparable or justified: Reviewer4 wanted an apples-to-apples comparison, Reviewer1 suggested the paper follow the evaluation paradigm used in earlier papers, and Reviewer2 described the absence of an explained baseline value.  Two reviewers (Reviewer4 and Reviewer2) described the lack of details on the parameters, architecture, and training regime used for the experiments.  The paper did not not justify which aspects of the modular system contributed to the observed performance (Reviewer4 and Reviewer1).   Several additional concerns were also raised.

The authors did not respond to any of the concerns raised by the reviewers.